# Acute chemical ingestion in the under 19 population in South Korea: A brief report

**Jae Hee Lee, Duk Hee Lee** *

Department of Emergency Medicine, Ewha Womans University, Seoul, South Korea

* ewhain78@gmail.com

## Abstract

### Background

Most people are frequently exposed to chemicals and chemical products. This study provides basic information on the outcomes of acute chemical ingestion of patients aged under 19 years.

### Methods

Patients aged under 19 years who had ingested chemicals and thus visited the emergency department between January 2011 and December 2016 were included in this study.

### Results

In all, 1,247 patients included (1,145 in the unintentional group and 102 in the intentional group). The mean age was 3.27±4.77 in the unintentional ingestion group and 16.49±1.94 in the intentional group. In the unintentional group, detergents were most frequently ingested (by 219 patients), followed by hypochlorite-based agents, ethanol, sodium hydroxide, acetone, silica gel, and citric acid. Cases of boric acid (odds ratio [OR] = 6.131), ethylene glycol (OR = 6.541), glacial acetic acid (OR = 7.644), other hydrocarbons (OR = 4.496), hypochlorite-based agent (OR = 2.627), nicotine (OR = 5.635), and sodium peroxocarbonate (OR = 6.783) ingestion was associated with a significantly high admission rate. In the intentional group, there were 54 cases of ingestion of hypochlorite-based agent, followed by detergent, ethylene glycol, ethanol, methanol and sodium peroxycarbonate. The significant risk factors for admission in the intentional group were ingestion of ethylene glycol (OR = 37.333) and hypochlorite-based agent (OR = 5.026). There was no mortality case.

### Conclusion

The most commonly ingested substances were sodium hypochlorite (hypochlorite-related agent), surfactants (detergent and soap), and ethanol. The ingestion of hypochlorite or ethylene glycol was the main risk factor for admission. Intentional ingestion was higher in adolescents than in children.

**Data Availability Statement:** The Korean Center for Disease Control (KCDC) is the authority for accessing the data analyzed, and there are ethical restrictions on sharing a dataset because the data contain potentially identifying information. The KCDC (http://www.cdc.go.kr) can be contacted for

data access via the Injury research team email (kcdcinjury@korea.kr) or by calling 82-43-719-7407. The authors used the dataset "Emergency Department-Based Injury In-depth Surveillance of the Korea Centers for Disease Control and Prevention (KCDC)." The authors did not have special access privileges.

**Funding:** This work was supported by the National Research Foundation of Korea (NRF) grant funded by the Korea government (MSIT) (2018R1C1B5046096), and a fund by Research of Korea Centers for Disease Control and Prevention (Emergency Department-Based Injury In-depth Surveillance). The funders had no role in study design, data collection and analysis, decision to publish, or preparation of the manuscript.

**Competing interests:** All authors have no potential conflicts of interest to disclose.

## Introduction

Up to 100,000 industrial chemicals are used each day in the United States [1]. Due to industrial development, humans are being increasingly exposed to chemicals [2]. Children are especially sensitive and vulnerable to exposed chemicals because their bodies are small and still developing [3]. In addition, their cognitive and behavioral processes are also developing, so there are many unintended and unrecognized effects of chemical poisonings [4, 5]. According to the 2017 National Poison Data System Annual Report from The American Association of Poison Control Centers (AAPCC), about 60% of all poisoning exposure occurred in children and young adults aged under 20 years. Among children under five years, unintentional exposure accounts for more than half of the exposure rate, and the substances that commonly cause poisoning are cosmetic and personal care products and household cleaning substances [6]. As such, children account for a large proportion of acute poisoning patients. However, prevalence and outcomes of chemical ingestion in South Korea are not known, since chemical exposure data system are usually focused on chemical accidents or industrial exposure, and there are no data on the exposure of pediatric patients to chemicals [7, 8]. Among the different type of chemical exposures, ingestion is more likely to cause systemic effects than skin or eye exposure. In this study, we analyzed the characteristics of patients under the age of 19 years who visited the emergency department (ED) for acute chemical ingestion to identify the patterns of chemical poisoning and improve the treatment strategies for such patients.

## Materials and methods

### Setting and data collection

This study retrospectively analyzed data from 2011 to 2016 recorded in the Emergency Department based Injury In-depth Surveillance that is monitored by the Korea Disease Control and Prevention Agency (KCDA) that was implemented in 2006 and involved 20 organizations from 2011 to 2014 and 23 organizations since 2015. The system collects data on epidemiology, treatment and outcomes of patients with injuries. Each hospital employs personnel responsible for data entry and quality control that are regularly trained and supervised by the KCDA.

Chemical ingestion was defined as the occurrence of injured patients who visited the ED with the mechanism of intoxication by artificial toxic substances. Toxic substances were classified according to the KCDA Toxic Substance Classification guidelines. This study involved patients aged < 19 years who met the criteria of chemical ingestion. Patients over 20 years of age and those with no information on intentionality or ED treatment results were excluded.

### Outcome measures

In the surveillance system, if the mechanism of injury is poisoning, the substance must be classified according to a specific code, and the general name or product name of the substance must be entered. In this study, two emergency physicians and one chemist identified and categorized the main components of the ingested chemicals. The incidence of chemical ingestions were classified into intentional and unintentional ingestion; the two groups were compared, and subgroup analysis was performed according to the types of chemicals reported. Data on the general characteristics, namely sex, age, mode of ED arrival, and the type of insurance were compared. The following data related to the injury were analyzed: the period from the time of injury to arrival at the ED, the place where the injury occurred, the activity being carried out at the time of injury, alcohol ingestion before the injury, date of the visit, time of the visit, and duration of the ED stay. To compare the severity of the patients (general ward [GW] admission or intensive care unit [ICU] care), data on treatment results and mortality were analyzed.

## Statistical analysis

We compared and analyzed the general characteristics and injury characteristics of the intentional and unintentional groups. The means and standard deviations are presented for continuous variables, and the number of patients and percentage are expressed for non-continuous variables. For items requiring statistical verification, an independent t-test was used for continuous variables and Chi-squared test or Fisher's exact test was used for non-continuous variables. Multiple logistic regression analysis was performed to determine the risk of hospital admission. Statistical analyses were performed using SPSS for Windows ver. 21 (IBM, Armonk, NY, USA). P values <0.05 were considered statistically significant.

## Compliance with ethical standards

The study complied with the Declaration of Helsinki, was approved by the institutional review board (IRB) of Ewha Womans' University Mok-dong hospital (IRB No.2019-08-006). The informed consent for gathering the data related ED visit was obtained from all participating patients or patients' parents/guardians by the KCDA according to National Research Committee.

## Results

From 2011 to 2016, 1,247 cases met the inclusion criteria of this study. The percentage of males was 55.8% and the mean age was 4.46 years in the total patients. There were 1,145 patients (91.8%) in the unintentional group and 102 cases (8.2%) in the intentional group.

Comparison of general characteristics of patients aged ≤19 in the intentional and unintentional groups, admitted to the emergency department for chemical ingestion during 2011–2016 (Table 1)

The mean patient age was 3.27±4.77 years in the unintentional group and 16.49±1.94 years in the intentional group. The proportion of females in the intentional group was 64.7%, which was higher than that in the unintentional group (42.4%). In the unintentional group, 90.7% of the ingestion incidents occurred at home. In the intentional group, most incidents occurred at residential and amusement facilities and cultural/public facilities. Regarding activity at the time of injury, ingestion in the unintentional group occurred during daily living (84.7%) and leisure (10.6%) activities. The proportion of alcohol ingestion before injury was 17.6% in the intentional group and 1.0% in the unintentional group. Patients in the unintentional group visited the ED mainly in the evening, and those in the intentional group, mainly at night. Further, 89.0% of the patients in the intentional group were discharged from the ED. In the intentional group, 53.9% were discharged from the ED, 35.3% were admitted to GW, and 10.8% were admitted to ICU. The mortality rate of the study sample was zero.

The top five chemicals ingested by patients aged <20 years in the unintentional and intentional ingestion groups at emergency department admission in South Korea, 2011–2016 (Table 2)

In the unintentional groups, detergents accounted for 219 of the 1,145 cases, followed by hypochlorite-related agents, ethanol, sodium hydroxide, and acetone. We analyzed the frequency of chemical agents by dividing the sample into three age groups: 0–5 years old, 6–12 years old, and 13–19 years. Nine hundred fifty-nine (83.8%) patients were aged under 5 years; in this group, the most commonly ingested substances were detergents, hypochlorite-based agents, and ethanol. Seventy-six patients were aged 6–12 years, and they primarily ingested hypochlorite-based agents and detergents. Finally, the third consisted of 110 patients, and the most commonly ingested substances were hypochlorite-based agents, ethanol, and detergents. A total of 126 patients in the unintentional group were admitted to the hospital, of which 36,

**Table 1. Comparison of general characteristics of patients aged ≤19 in the intentional and unintentional groups, admitted to the emergency department for chemical ingestion during 2011–2016.**

| | Unintentional | | Intentional | | Total | | p-value |
|---|---|---|---|---|---|---|---|
| | N | % | N | % | N | % | |
| No. of patients | 1145 | 91.8% | 102 | 8.2% | 1247 | 100% | |
| Sex | | | | | | | <0.000 |
| Male | 660 | 57.6% | 36 | 35.3% | 696 | 55.8% | |
| Female | 485 | 42.4% | 66 | 64.7% | 551 | 44.2% | |
| Age (yrs, mean±SD) | 3.27 | (4.77) | 16.49 | (1.94) | 4.46 | (5.95) | <0.000 |
| Mode of arrival | | | | | | | 0.713* |
| Walk-in (car, by foot, etc.) | 853 | 74.5% | 81 | 79.4% | 934 | 74.9% | |
| 119 | 173 | 15.1% | 12 | 11.8% | 185 | 14.8% | |
| Private ambulance | 116 | 10.1% | 9 | 8.8% | 125 | 10.0% | |
| Others | 2 | 0.2% | 0 | 0.0% | 2 | 0.2% | |
| Unknown | 1 | 0.1% | 0 | 0.0% | 1 | 0.1% | |
| Time interval from injury to ED visit (h) | 9.59 | (37.23) | 14.12 | (43.17) | 9.87 | (37.54) | 0.246 |
| Insurance | | | | | | | 0.765* |
| National health insurance | 949 | 82.9% | 87 | 85.3% | 1036 | 83.1% | |
| Vehicle | 159 | 13.9% | 13 | 12.7% | 172 | 13.8% | |
| Medicaid beneficiary | 17 | 1.5% | 2 | 2.0% | 19 | 1.5% | |
| Self-pay (uninsured) | 17 | 1.5% | 0 | 0.0% | 17 | 1.4% | |
| Others | 3 | 0.3% | 0 | 0.0% | 3 | 0.2% | |
| Place | | | | | | | 0.005* |
| House | 1039 | 90.7% | 83 | 81.4% | 1122 | 90.0% | |
| Residential facility | 7 | 0.6% | 3 | 2.9% | 10 | 0.8% | |
| School, education facility | 32 | 2.8% | 5 | 4.9% | 37 | 3.0% | |
| Amusement, cultural public facility | 7 | 0.6% | 3 | 2.9% | 10 | 0.8% | |
| Commercial facility | 30 | 2.6% | 2 | 2.0% | 32 | 2.6% | |
| | 11 | 1.0% | 3 | 2.9% | 14 | 1.1% | |
| Other† | 15 | 1.3% | 2 | 2.0% | 17 | 1.4% | |
| Unknown | 4 | 0.3% | 1 | 1.0% | 5 | 0.4% | |
| Activity | | | | | | | <0.000* |
| Work | 8 | 0.7% | 0 | 0.0% | 8 | 0.6% | |
| Unpaid labor | 6 | 0.5% | 0 | 0.0% | 6 | 0.5% | |
| Education | 23 | 2.0% | 0 | 0.0% | 23 | 1.8% | |
| Leisure | 121 | 10.6% | 0 | 0.0% | 121 | 9.7% | |
| Daily living activity | 970 | 84.7% | 1 | 1.0% | 971 | 77.9% | |
| Other‡ | 14 | 1.2% | 101 | 99.0% | 115 | 9.2% | |
| Unknown | 3 | 0.3% | 0 | 0.0% | 3 | 0.2% | |
| Alcohol ingestion before injury | | | | | | | <0.000* |
| No | 1120 | 97.8% | 73 | 71.6% | 1193 | 95.7% | |
| Yes | 12 | 1.0% | 18 | 17.6% | 30 | 2.4% | |
| Unknown | 13 | 1.1% | 11 | 10.8% | 24 | 1.9% | |
| Day of presentation | | | | | | | 0.281 |
| Weekday (Mon-Thu) | 644 | 56.2% | 63 | 61.8% | 707 | 56.7% | |
| Weekend (Fri-Sun) | 501 | 43.8% | 39 | 38.2% | 540 | 43.3% | |
| Time of presentation | | | | | | | <0.000 |
| Day | 351 | 30.7% | 27 | 26.5% | 378 | 30.3% | |
| Evening | 671 | 58.6% | 41 | 40.2% | 712 | 57.1% | |

*(Continued)*

**Table 1.** (Continued)

| | Unintentional | | Intentional | | Total | | p-value |
|---|---|---|---|---|---|---|---|
| | N | % | N | % | N | % | |
| Night | 123 | 10.7% | 34 | 33.3% | 157 | 12.6% | |
| ED stay (h) | 29.69 | (62.68) | 16.71 | (6.84) | 28.46 | (59.79) | <0.000 |
| ED treatment result | | | | | | | <0.000 |
| Discharge | 1019 | 89.0% | 55 | 53.9% | 1074 | 86.1% | |
| General ward | 123 | 10.7% | 36 | 35.3% | 159 | 12.8% | |
| Intensive care unit | 3 | 0.3% | 11 | 10.8% | 14 | 1.1% | |
| Hospital mortality | 0 | 0.0% | 0 | 0.0% | 0 | 0.0% | |

Quantitative data are expressed as mean (standard deviation), and categorical data are presented as number of subjects (%). Independent t-test was used for continuous variable analysis, while Chi-squared test or Fisher's exact test* was used for categorical variable analysis. Other† includes medical facilities, sports facilities, roads, transportation areas except roads, factories, and industrial facilities. Other‡ includes exercise, hospital treatment, and travel.

16, and 6 had ingested hypochlorite-based agents, detergents, and ethanol or acetone, respectively.

In the intentional group, 54 patients had ingested hypochlorite-based agents; the others had ingested detergents, ethylene glycol, ethanol, and methanol. On dividing the ages into three groups, there were no patients in the group of patients aged under 5 years, four in the 6–12 years group, and 98 (96%) in the 13–19 years group; 47 patients were admitted to the ICU: 28 had ingested hypochlorite-based agents, eight, ethylene glycol, two, detergent, and two, ethanol.

Unintentional chemical ingestion in patients (<20 years) in the discharge and admission subgroups, South Korea, 2011–2016 (Tables 3 and 4)

**Table 2.** The top five chemicals ingested by patients aged <20 years in the unintentional and intentional ingestion groups at emergency department admission in South Korea, 2011–2016.

| Chemical | Total N | Age group | | | admission |
|---|---|---|---|---|---|
| | | ~5 yrs | 6~12yrs | 13~19yrs | |
| Unintentional | | | | | |
| Number of cases | 1145 | 959 | 76 | 110 | 126 |
| Top 5 agent | | | | | |
| Detergents/soaps-anionic and nonionic | 219 | 192 (87.7) | 10 (4.6) | 17 (7.8) | 16 (7.3) |
| Hypochlorite-based agents | 195 | 143 (73.3) | 23 (11.8) | 29 (14.9) | 36 (18.5) |
| Ethanol | 137 | 116 (84.7) | 1 (0.7) | 20 (14.6) | 6 (4.4) |
| Sodium hydroxide | 65 | 57 (87.7) | 6 (9.2) | 2 (3.1) | 5 (7.7) |
| Acetone | 57 | 52 (91.2) | 3 (5.3) | 2 (3.5) | 6 (10.5) |
| Intentional | | | | | |
| Number of cases | 102 | 0 | 4 | 98 | 47 |
| Top 5 agent | | | | | |
| Hypochlorite-based agents | 54 | 0 (0.0) | 1 (1.9) | 53 (98.1) | 28 (51.9) |
| Detergents/soaps-anionic and nonionic | 10 | 0 (0.0) | 0 (0.0) | 10 (100.0) | 2 (20.0) |
| Ethylene glycol | 9 | 0 (0.0) | 0 (0.0) | 9 (100.0) | 8 (88.9) |
| Ethanol | 4 | 0 (0.0) | 0 (0.0) | 4 (100.0) | 2 (50.0) |
| Methanol | 4 | 0 (0.0) | 0 (0.0) | 4 (100.0) | 1 (25.0) |

Data are presented as number of subjects (percentages).

**Table 3. General characteristics of unintentional chemical ingestion in patients (<20 years) in the discharge and admission subgroups, South Korea, 2011–2016.**

| | Discharge | | Admission | | Total | | p-value |
|---|---|---|---|---|---|---|---|
| | N | % | N | % | N | % | |
| No. of patients | 1019 | 89% | 126 | 11% | 1145 | 100% | |
| Sex | | | | | | | |
| Male | 583 | 57.2% | 77 | 61.1% | 660 | 57.6% | 0.403 |
| Female | 436 | 42.8% | 49 | 38.9% | 485 | 42.4% | |
| Age (yrs, mean±SD) | 3.06 | (4.54) | 4.98 | (6.06) | 3.27 | (4.77) | 0.001 |
| Time interval from injury to ED visit (h) | 9.90 | (38.83) | 7.05 | (19.92) | 9.59 | (37.23) | 0.417 |
| Place | | | | | | | 0.520* |
| House | 923 | 90.6% | 116 | 92.1% | 1039 | 90.7% | |
| Residential facility | 5 | 0.5% | 2 | 1.6% | 7 | 0.6% | |
| School, education facility | 27 | 2.6% | 5 | 4.0% | 32 | 2.8% | |
| Amusement, cultural public facility | 7 | 0.7% | 0 | 0.0% | 7 | 0.6% | |
| Commercial facility | 28 | 2.7% | 2 | 1.6% | 30 | 2.6% | |
| Outdoor, river, sea | 10 | 1.0% | 1 | 0.8% | 11 | 1.0% | |
| Other† | 15 | 1.5% | 0 | 0.0% | 15 | 1.3% | |
| Unknown | 4 | 0.4% | 0 | 0.0% | 4 | 0.3% | |
| Activity | | | | | | | 0.102* |
| Work | 8 | 0.8% | 0 | 0.0% | 8 | 0.7% | |
| Unpaid labor | 5 | 0.5% | 1 | 0.8% | 6 | 0.5% | |
| Education | 21 | 2.1% | 2 | 1.6% | 23 | 2.0% | |
| Leisure | 113 | 11.1% | 8 | 6.3% | 121 | 10.6% | |
| Daily living activity | 860 | 84.4% | 110 | 87.3% | 970 | 84.7% | |
| Other‡ | 10 | 1.0% | 4 | 3.2% | 14 | 1.2% | |
| Unknown | 2 | 0.2% | 1 | 0.8% | 3 | 0.3% | |
| ED stay (h) | 31.98 | (66.09) | 11.23 | (0.61) | 29.69 | (62.68) | <0.000 |

Quantitative data are expressed as mean (standard deviation), and categorical data are presented as number of subjects (%). Independent t-test was used for continuous variable analysis, while the chi-squared test or Fisher's exact test* for categorical variable analysis. Other† included medical facilities, sports facilities, roads, transportation areas except roads, and factories and industrial facilities. Other‡ included exercise, hospital treatment, and travel.

A total of 126 patients in the unintentional group were admitted to the ED: 36, 16, and six patients had ingested hypochlorite-based agents, detergent, and ethanol or acetone, respectively. The characteristics of the admitted and discharged patients in the unintentional group were compared. The mean age of admitted patients was 4.98 years, which was significantly

**Table 4. Multivariate analysis of unintentional chemical ingestion in patients (<20 years) in the admission subgroup.**

| | Exp(B) | 95% C.I. for EXP(B) | | Sig. |
|---|---|---|---|---|
| | | Lower | Upper | |
| Age | 1.068 | 1.032 | 1.105 | <0.000 |
| Boric acid | 6.131 | 1.852 | 20.293 | 0.003 |
| Ethylene glycol | 6.541 | 1.008 | 42.446 | 0.049 |
| Glacial acetic acid | 7.644 | 2.027 | 28.822 | 0.003 |
| Other hydrocarbon | 4.496 | 1.366 | 14.801 | 0.013 |
| Hypochlorite-based agents | 2.627 | 1.663 | 4.152 | <0.000 |
| Nicotine | 5.635 | 2.655 | 11.959 | <0.000 |
| Sodium peroxocarbonate | 6.783 | 2.421 | 19.002 | <0.000 |

higher than that of the discharge group (3.06 years). In addition, there were no significant differences in sex, place of poisoning, activity, alcohol ingestion, and time of presentation, which were significantly different between the admitted and discharged patients (Table 3). Regression analysis was performed to identify the risk factors for admission in the unintentional groups. As shown in Table 3, the variables that showed significant differences in univariate analysis were age and duration of ED stay. However, ED stay was excluded because it was not a factor that contributed to the severity of the patient. Therefore, in the multiple logistic regression, age and ingested chemicals were selected and implemented, and the significant variables obtained by backward stage selection (likelihood ratio) are presented in Table 4. Boric acid (OR = 6.131), ethylene glycol (OR = 6.541), glacial acetic acid (OR = 7. 644), other hydrocarbons (OR = 4.496), hypochlorite related agents (OR = 2.627), nicotine (OR = 5.635), and sodium peroxycarbonate (OR = 6.783) ingestion significantly increased the admission rate. (Table 4)

## Discussion

Considering the high rate of industrialization, numerous chemicals are being produced and distributed, resulting in increasing intentional and unintentional exposure to harmful substances [9]. To prevent exposure to lethal chemicals, the current exposure situation must be analyzed. Data on chemical exposure in industrial working environments or due to accidents are collected and reported in various government offices. However, there are no data on chemical exposure in everyday life in South Korea [10].

Almost all patients complaining of acute chemical exposure are admitted to the ED, but people facing chronic exposure to chemicals because of unsafe work environments may not visit the hospital. In the United States, detailed reporting on exposure and poisoning, including drugs and chemicals, is performed annually based on monitoring standards for toxic substances [3, 6]. These data have been used to conduct studies on chemical poisoning [11]. In South Korea, no institution or organization monitors exposure or treatment for poisoning. The ED-based injury surveillance (including intoxication) system is the only available nationwide data can be used for epidemiological investigations and clinical analysis of acute poisoning.

In this study, we analyzed the data of patients aged under 19 years who visited the ED for chemical ingestion. The most frequently ingested chemicals were sodium hypochlorite (hypochlorite-based agents), surfactants (detergents and soap), and ethanol. The most frequently ingested chemicals requiring admission were sodium hypochlorite, surfactant, nicotine, and ethylene glycol. A total of 91.8% of all chemical ingestions were unintentional. Analysis of the unintentional ingestion stratified by age showed that 83.6% of the patients were aged under 5 years. The overall rate intentional ingestion accounted for 8.2%. In the 13-18-year-old-group, 96.1% of the cases were intentional ingestion cases. The 6-12- year-old group had the lowest frequency of total chemical ingestion (6.4%). (S1 Table)

Children aged 1–5 years become increasingly mobile with age; this enables their exploratory behavior and increases their accessibility of different objects. However, since children in this age group are unable to make accurate decisions, many incidents of unintentional/non-food poisoning occur [12]. Children between 6 and 12 years accounted for the lowest proportion of childhood poisoning cases. The incidence of intentional poisoning, especially suicide attempts, is found to be high in the population aged above 13 years as is the rate of addiction [4, 11].

In this study, sodium hypochlorite ingestion accounted for the highest percentage of admitted patients and was a risk factor for admission in both the intentional and unintentional ingestion groups. Hypochlorite agents are commonly used at home for bleaching, discoloring, and mold removal. Low concentrations of 3% - 5% hypochlorite agents used for household

purpose can irritate the skin but do not cause serious burns. Industrial hypochlorite agents have a concentration of 20% hypochlorite and can cause corrosive injury and pulmonary irritation [13]. In this study, hypochlorite ingestion was associated with a high rate of hospitalization but did not cause fatal damage leading to death, possibly because the majority of the hypochlorite agents ingested were domestic-use agents, given that 90% of the poisonings incidents had occurred at home.

Detergent and soap ingestion accounted for the second highest proportion of total and admitted patients. Soap and liquid detergents are known to cause little or no fatal damage in small doses. However, when ingested, they may cause nausea and vomiting, so symptomatic treatment may be necessary, depending on the severity [14].

Ethylene glycol ingestion was a risk factor for admission in the unintentional group. Ethylene glycol is a colorless, odorless liquid and is widely used as a car antifreeze solution. Ethylene glycol has toxic effects resulting from the metabolite produced by alcohol dehydrogenase, and patients who ingest ethylene glycol have a serious prognosis. Metabolic acidosis and kidney injury can occur. Antidote (ethanol and fomepizole) administration and dialysis should be considered for these patients [15, 16].

A limitation of this study was that the types of chemicals were not specifically classified in the data collection phase. Since the initial data only recorded the general chemical names, the components were identified based on the product names entered in the system, and reclassification was needed. It is advisable to collect detailed information in the data collection stage. In addition, the absence of information on the amount and concentration of the chemicals further limited the data analysis process.

## Conclusion

According to the results, the most commonly ingested chemicals were sodium hypochlorite (hypochlorite-related agents), surfactants (detergents and soap), and ethanol. The overall mortality in our sample was zero. Hypochlorite and ethylene glycol ingestion were risk factors for admission. The incidence of intentional ingestion was higher in adolescents than in children. This is a useful resource for the prevention and management of chemical poisoning and for anticipatory guidance on chemical ingestion in pediatric and adolescent populations.

## Supporting information

**S1 Table. Total incidence and admission of chemical ingestion in children at ED by age group, South Korea, 2011–2016.**
(DOCX)

## Author Contributions

**Conceptualization:** Duk Hee Lee.

**Data curation:** Jae Hee Lee, Duk Hee Lee.

**Formal analysis:** Duk Hee Lee.

**Funding acquisition:** Duk Hee Lee.

**Visualization:** Jae Hee Lee.

**Writing – original draft:** Jae Hee Lee, Duk Hee Lee.

**Writing – review & editing:** Jae Hee Lee, Duk Hee Lee.

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
