## [Decision Letter · Decision Letter 0]

6 Oct 2020

PONE-D-20-27951

Clinical outcomes of acute chemical ingestions in pediatrics and adolescents admitted to Emergency department, South Korea

PLOS ONE

Dear Dr. Duk Hee Lee,

Thank you for submitting your manuscript to PLOS ONE. After careful consideration, we feel that it has merit but does not fully meet PLOS ONE’s publication criteria as it currently stands. Therefore, we invite you to submit a revised version of the manuscript that addresses the points raised during the review process.

After a statistician analysis we conclude that the sample size is too small especially for intentional group. Some crucial factors were not measured or reported.  We recommend the authors to simplify the analysis and re-write the manuscript as a brief report.

Please submit your revised manuscript when you are ready, then submit your revision, log on to https://www.editorialmanager.com/pone/ and select the 'Submissions Needing Revision' folder to locate your  manuscript file.

We look forward to receiving your revised manuscript.

Kind regards,

Martina Crivellari

Academic Editor

PLOS ONE

Journal Requirements:

Reviewers' comments:

Reviewer's Responses to Questions

**Comments to the Author**

1. Is the manuscript technically sound, and do the data support the conclusions?

Reviewer #1: Partly

2. Has the statistical analysis been performed appropriately and rigorously? 

Reviewer #1: No

3. Have the authors made all data underlying the findings in their manuscript fully available?

Reviewer #1: Yes

4. Is the manuscript presented in an intelligible fashion and written in standard English?

Reviewer #1: Yes

5. Review Comments to the Author

Reviewer #1: The retrospective study reported the major types of chemical poisoning agents in pediatrics patients who visited ED in South Korea from 2011 to 2016 and accessed the associations between hospital admission and potential risk factors. The authors published a similar study for adult patients using the same database, study design and analytic methods.

The sample size of intentional group is too small to be analyzed using multiple logistic regression with so many dummy covariates. Most of chemical ingestions cannot be included in analysis due to a lack of cases. The result of multivariable analysis of intentional chemical ingestion for admission is bias.

I recommend to simplify the analysis and re-write the manuscript as a brief report.

For pediatrics especially for young age kids, exposure concentrations / doses and period from time of arrival to ED admission are crucial factors. However, the authors didn’t collect and report the corresponding data. It may bring significant bias by ignoring those two factors.

Are there any clinical/statistical reasons to divide the ages into three groups? It makes the tables too complicated and may dilute the primary take home message.

For categorical characteristics with more than two categories, it is suspected and tedious to report p-values of Chi-square tests of each category without multiple adjustment. I recommend to perform one single Chi-square test for categorical characteristics and report a single p-value.

Almost all patients in intentional group reported “Other” for activity (101/102). Is there free text clarification available in database?

Reporting total in tables is tedious. Also, for sex and day of presentation, reporting one category is good enough.

The author didn’t clearly illustrate covariates and the corresponding variable selection method used in multiple logistic regression.

6. PLOS authors have the option to publish the peer review history of their article (what does this mean?). If published, this will include your full peer review and any attached files.

Reviewer #1: No

---

## [Author Response · Author response to Decision Letter 0]

27 Oct 2020

I appreciate your careful reading and proper comments. 

Reviewer #1: The retrospective study reported the major types of chemical poisoning agents in pediatrics patients who visited ED in South Korea from 2011 to 2016 and accessed the associations between hospital admission and potential risk factors. The authors published a similar study for adult patients using the same database, study design and analytic methods. 

The sample size of intentional group is too small to be analyzed using multiple logistic regression with so many dummy covariates. Most of chemical ingestions cannot be included in analysis due to a lack of cases. The result of multivariable analysis of intentional chemical ingestion for admission is bias.

I recommend to simplify the analysis and re-write the manuscript as a brief report. 

RESPONSE: We agree that multiple logistic regression is not appropriate because the sample size of the intentional group is small. After analyzing the study population by dividing it into intentional and unintentional ingestion groups, subgroup analysis was conducted only for the unintentional group. The analysis was simplified and edited in the form of a brief report.

For pediatrics especially for young age kids, exposure concentrations / doses and period from time of arrival to ED admission are crucial factors. However, the authors didn’t collect and report the corresponding data. It may bring significant bias by ignoring those two factors. 

RESPONSE: The lack of information on the concentration and amount of chemicals is a clear limitation of this study, but I think it makes sense to understand the common types of chemicals ingested by pediatric patients. The period from arrival to admission to ED is indicated in the table as “Time Interval from Injury to ED Visit” (Tables 1 and 3), and there was no significant difference between the discharge and admitted subgroups in the unintended ingestion group.

Are there any clinical/statistical reasons to divide the ages into three groups? It makes the tables too complicated and may dilute the primary take home message.

RESPONSE: We agree with your opinion that the classification is difficult to understand. However, among children, the characteristics of the preschool age, school age, and adolescent age are very different. The same criteria have been used in the “2017 Annual Report of the American Association of Poison Control Centers’ National Poison Data System (NPDS): 35th Annual Report” to analyze the distribution by age group. 

For categorical characteristics with more than two categories, it is suspected and tedious to report p-values of Chi-square tests of each category without multiple adjustment. I recommend to perform one single Chi-square test for categorical characteristics and report a single p-value.

RESPONSE: The analysis was re-run to derive a single p-value, and Fisher's exact test was used when the expected count for less than 5 cells was 25% or more.

Almost all patients in the intentional group reported “Other” for activity (101/102). Is there free text clarification available in the database?

RESPONSE: The free text clarification is summarized as follows: 

Statistics

Activity_others

N Valid 102

 Missing 0

Activity_others

 Frequency Percent Valid Percent Cumulative Percent

Valid 1 1.0 1.0 1.0

 Toxic ingestion 2 2.0 2.0 2.9

 Suicide 21 20.6 20.6 23.5

 Suicide/self-harm 1 1.0 1.0 24.5

 For suicide 3 2.9 2.9 27.5

 Suicidal attempt 35 34.3 34.3 61.8

 Suicidal attempt/self-harm 1 1.0 1.0 62.7

 Self-harm 13 12.7 12.7 75.5

 Self-harm or suicide 2 2.0 2.0 77.5

 Attempt of self-harm 1 1.0 1.0 78.4

 Self-harm, suicide 3 2.9 2.9 81.4

 Self-harm/suicide 1 1.0 1.0 82.4

 Self-harm suicide 18 17.6 17.6 100.0

 Total 102 100.0 100.0 

 

Reporting total in tables is tedious. Also, for sex and day of presentation, reporting one category is good enough.

RESPONSE: Reflecting your point of view, we have categorized the items with a frequency less than 0.5% as “others” and added a description at the bottom of the table.

The author didn’t clearly illustrate covariates and the corresponding variable selection method used in multiple logistic regression.

RESPONSE: As shown in Table 3, among the variables collected in this study, the variables that showed significant differences in the univariate analysis were age and ED stay. However, ED stay was excluded because it was not a factor that contributed to the severity of the patient. Therefore, in the multiple logistic regression, age and ingested chemicals were selected and implemented, and significant variables obtained by backward stage selection (likelihood ratio) are presented in Table 4. This explanation is added to the text.

---

## [Editor Report · Decision Letter 1]

3 Nov 2020

Acute chemical ingestion in the under 19 population in South Korea: a brief report

PONE-D-20-27951R1

Dear Dr. Hee Lee,

We’re pleased to inform you that your manuscript has been judged scientifically suitable for publication and will be formally accepted for publication once it meets all outstanding technical requirements.

Kind regards,

Martina Crivellari

Academic Editor

PLOS ONE
---

## [Editor Report · Acceptance letter]

5 Nov 2020

PONE-D-20-27951R1 

Acute chemical ingestion in the under 19 population in South Korea: a brief report 

Dear Dr. Lee:

I'm pleased to inform you that your manuscript has been deemed suitable for publication in PLOS ONE. Congratulations! Your manuscript is now with our production department. 

Kind regards, 

on behalf of

Dr. Martina Crivellari 

Academic Editor

PLOS ONE